# Selective Laser Melting of CuSn10: Simulation of Mechanical Properties, Microstructure, and Residual Stresses

**DOI:** 10.3390/ma15113902

**Published:** 2022-05-30

**Authors:** Robert Kremer, Somayeh Khani, Tamara Appel, Heinz Palkowski, Farzad Foadian

**Affiliations:** 1Faculty of Mechanical Engineering, Dortmund University of Applied Sciences and Arts, Sonnenstr. 96, 44139 Dortmund, Germany; robert.kremer@fh-dortmund.de (R.K.); tamara.appel@fh-dortmund.de (T.A.); farzad.foadian@fh-dortmund.de (F.F.); 2Institute of Metallurgy, Clausthal University of Technology, Robert-Koch-Strasse 42, 38678 Clausthal-Zellerfeld, Germany; somayeh.khani@tu-clausthal.de

**Keywords:** selective laser melting, CuSn10, residual stresses, microstructure, FEM simulation, Ansys

## Abstract

In this study, the evolution of mechanical properties, microstructure, and residual stresses during selective laser melting of CuSn10 components was studied. To provide a proper material model for the simulations, various CuSn10 parts were manufactured using selective laser melting and examined. The manufactured parts were also used to validate the developed model. Subsequently, a sequentially coupled thermal–mechanical FEM model was developed using the Ansys software package. The developed model was able to deliver the mechanical properties, residual stresses, and microstructure of the additively manufactured components. Due to introducing some simplifications to the model, a calibration factor was applied to adjust the simulation results. However, the developed model was validated and showed a good agreement with the experimental results, such as measured residual stresses using the hole drilling method, as well as mechanical properties of manufactured parts. Moreover, the developed material model was used to simulate the microstructure of manufactured CuSn10. A fine-grain microstructure with an average diameter of 19 ± 11 μm and preferred orientation in the Z-direction, which was the assembly direction, was obtained.

## 1. Introduction

Selective laser melting (SLM) is an additive manufacturing process from the powder-bed-based beam melting process group. Manufacturing occurs by applying fine powder coatings in layers and then melting them locally using a moving laser beam. In the process, melt tracks are created in the powder bed, which form the individual layers via overlapping. The desired component is created layer by layer through a repeating cycle of powder application and layer exposure. Due to the layer-by-layer and tool-less production, component complexity and production effort are decoupled from each other [1,2,3].

The biggest challenge in introducing SLM technology into new areas is the qualification and certification of the manufactured parts [4]. A part must be manufactured without defects in a repeatable manner for qualification [5]. In this regard, the mechanical properties must be within a defined specification so that SLM fabricated parts can perform their intended task. In SLM manufacturing, the high temperature gradients during local melting and layer-by-layer manufacturing result in strong residual stresses in the component and a preferential direction in the microstructure (anisotropy) [6,7,8,9,10,11]. This can significantly reduce the usability of the manufactured parts. In particular, since the measurement of residual stresses and anisotropy is time-consuming and costly, and these measurements always cover only small areas, they pose a challenge [12]. In addition, the behavior of these two quantities during the buildup process and component deformation cannot be determined by measurements. Using suitable simulation methods makes it possible to completely map the described variables in the entire component, which means that the challenges of SLM manufacturing can be better met [13,14,15,16,17].

Chengyang Deng et al. [18] was able to process CuSn10 using a low laser power of 100 W despite the high reflection of CuSn10. Condruz (2020) [12] found a significantly different material behavior depending on the orientation to the buildup direction.

Javed Akram et al. [19] studied the influence of microstructure on the SLM process. They showed that with the help of simulation, it is possible to simulate the microstructure in the SLM process in a suitable way. He also used a 2D approach to simulate the anisotropy and grain size for the SLM process. Keller [20] described a simulation setup in which the melting of the individual melt tracks was simulated, providing very accurate information about the stress state in the manufactured material but requiring an immense amount of computation.

In most material simulations, simplifications were made to the approach that reduced the imaging capability of the simulation but immensely diminished the computational cost [21].

Maiwald et al. investigated the SLM process using simulations in Ansys. In order to validate the simulation results, they used a correction factor to compensate for simplifications that had been made [22].

This study is part of a larger project on Integrated Computational Materials Engineering (ICME) in additive manufacturing [23]. The goal was to combine simulation methods and models from the electronic to the structural scale to simulate the texture and residual stresses of different materials in SLM. The focus was on the resulting residual stresses and anisotropy due to manufacturing. This approach is similar to that of Foadian et al., who studied various materials using ICME in the tube-drawing process [24,25]. In this work, CuSn10 was used, which contains two main elements that offer more complexity in ICME simulations compared with pure metals. In the long term, the model created will be extended to multi-material alloys. This paper presents the processing and parameterization of the CuSn10 alloy, process simulation, and microstructure prediction.

Figure 1 shows the approach chosen for the investigations. Initially, the CuSn10 powder was parameterized on the existing SLM system, where tensile samples were produced to determine the mechanical–technological characteristic values. This provided the manufacturing parameters required for the simulations, such as the laser beam power or scanning speed, as well as the necessary mechanical characteristic values. The required thermal material data were taken from the literature. These parameters were combined into a material model and used for the process simulations, as well as the microstructure simulation. The process simulation data was used to optimize the distortion and residual stress during production. However, the microstructure simulation was used to estimate the anisotropy in the component, which was considered in the design process. In the medium term, both simulations are to be linked via the Hall–Petch relationship. In the long term, the findings of this investigation will be used for the overall ICME project.

## 2. Materials and Methods

Gas-atomized CuSn10 powder was used for manufacturing the SLM products. SEM investigations were used to determine the particle size and morphology, which showed the powder’s spherical shape and good morphology in the as-received powder. A total of 1068 particles were measured for the size distribution study. The average particle size of 17.3 ± 7.3 μm was determined. The total grain size distribution is shown in Figure 2. In addition, the flowability was investigated using flow analysis, where a flow time of 7.1 ± 0.3 s was measured.

Furthermore, cross-sections of the powder were examined, and the microstructure was studied. For this purpose, the powder was hot-embedded in a graphite-containing epoxy resin and metallographically processed. Images of the powder and the microstructure can be seen in Figure 3.

In this work, the *MLab R* SLM equipment from Concept Laser was used. The parameters that were used for manufacturing the samples are shown in Table 1. These were determined experimentally in advance. The manufactured test components were examined metallographically and hydrostatically for their density. In addition, tensile tests and hardness measurements were carried out. For the tensile test, flat tensile specimens that satisfied DIN EN ISO 6892-1 were selected and manufactured in the Z-direction and reworked via grinding. An etchant consisting of deionized water, ammonia, and hydrogen peroxide was used for the metallographic examination.

For the residual stress measurements, a component susceptible to residual stress was needed that was easy to measure and safe to assemble. Following the VDI guideline 3405 [1], a cantilever beam with a solid support structure was designed, which is shown schematically in Figure 4. Due to the large area parallel to the base area, high residual stresses were expected. Five of these so-called distortion bridges were manufactured. One was used to measure the residual stresses at the measuring points MP1, MP2, and MP3. For this purpose, the residual stresses were measured using the hole-drilling method with up to a 1.2 mm depth and a measuring interval of 0.1 mm. The support structures were cut through in the remaining deformation bridges, causing distortion resulting from the residual stresses. This was determined along the measuring section X1 using a tactile measuring method. Only minor deviations between the individual components were determined. Figure 5 shows one of the fabricated distortion bridges on the build plate.

A material model was developed using the investigated mechanical properties. Some main parameters are given in Table 2. The parameters were partly measured and partly taken from the literature. The aim was to map the temperature dependency of the parameters over the entire range occurring in the buildup process. This was partly achieved through data from the German Copper Institute and a research paper by G. Branner on the modeling of transient effects in the structural simulation of layer construction processes with copper alloys. The two sources were used because they provide temperature-dependent data [26,27]. The relationship between stress and strain is represented by the bilinear isotropic strain hardening material model. This was considered sufficient as it could represent the expected deformations. The process simulation was developed in Ansys Workbench 2020 R2 as a sequentially coupled thermal–mechanical simulation using the Additive Wizard [28].

Some simplifications were introduced to reduce the required computational time. For example, the laser–material interaction was replaced by a thermal boundary condition in which the melting temperature was applied to the surface. Thus, the motion after the melt solidifies was simulated, and the actual melting process was neglected. It was assumed that the melting process was without any defects since defects during melting cannot be considered in the simulation. The temperature boundary condition was applied to the entire layer simultaneously so that residual stresses between the two layers could be represented. In addition, multiple component layers were always grouped since similar behavior of adjacent component layers was assumed. A thermal boundary condition replaced the surrounding powder. The material increase during manufacturing was enabled by the activation or deactivation of finite elements. The elements of the layers not yet manufactured were deactivated and only activated when the corresponding layer was manufactured. The simplifications and the comparison between the real and the simulated SLM process is shown in Figure 6. To compensate for the inaccuracies caused by the simplifications, an experimentally determined correction factor was used.

The thermal boundary condition and the time for powder coating required in the thermal simulation are presented in Table 3.

For the calibration of the simulation, a process simulation of the deformation bridge was set up and built without correction (correction factor = 1). A linear approach with a Cartesian method and an element size of 0.1 mm was used for meshing. The distortion that occurred in the simulation over the coordinate X1 was matched with the experimental tests (see Figure 4). A correction factor was introduced to compensate for inaccuracies due to the simplifications and deviations in the material model that affected the magnitude of the resulting deformation. For the iterations of the correction factor, a deviation of 1% at the end of the measuring section X1 was selected as a termination criterion. Once this limit was reached, the simulation was considered calibrated and used to calculate the residual stresses at three measuring points: MP1, MP2, and MP3.

To simulate the texture, the microstructure tool of Ansys *Additive Science 2021 R1* was used, which creates a 2D cellular automaton to simulate the grain morphology in the SLM process. For this purpose, the simulation domain is divided into finite cells with state-described variables. These contain, among other things, information on the growth direction and the solid content. By applying deterministic or probabilistic transformation rules, the spatial and temporal development is determined by the state of its neighboring cells described [29]. The nucleation law [30], which takes into account the effects of supercooling and the cooling rate, is used as the starting point. Once a new nucleus has emerged, the preferred direction of growth is calculated based on the normal angle between the nucleus and the moving heat source. Thermal gradients and cooling rates are kept constant in the calculation. Furthermore, constitutional undercooling is neglected. As a result, information about grain size, grain shape, and texture can be calculated [19].

A thermal simulation was first performed in Ansys for the microstructure simulation to determine the cooling rate and temperature gradient. The simulation was carried out with the geometric data of the melt track, which was determined using metallographic investigations. Table 4 shows the input values used in the microstructure simulation.

## 3. Results and Discussion

The density of the components was measured hydrostatically on six samples. A relative and absolute density of 99.77 ± 0.21% and 8.759 ± 0.018 g/cm³ were determined, respectively. In addition, the density was checked metallographically to obtain a better picture of the pore design. Tensile tests, hardness measurements, and microstructural examinations were carried out to examine the produced material. As shown in Table 5, the additively prepared specimens had a yield strength of 420 ± 14 MPa, an ultimate tensile strength of 487 ± 12 MPa, and an elongation at break of 5 ± 0.5%. The hardness measurements resulted in a hardness of 173 ± 3 HV30.

The stress–strain diagrams are shown in Figure 7. Almost no necking was observed, which resulted in no drop after the ultimate tensile stress was reached.

SEM images of a sample after the SLM process are presented in Figure 8. The transverse and longitudinal sections are shown in Figure 8a,b, respectively. The individual melt traces can be seen, which were only slightly deep in the longitudinal section.

Figure 9 indicates the deformation in the buildup direction along the measuring section X1. The dashed graph presents the average deformation of the measured specimens after removing the support material. The solid line shows the simulated deformation without correction (correction factor = 1). As in [22], the deformation was significantly higher than in the laboratory tests. This was assumed to be related to an insufficiently accurate material model and the introduced simplifications, especially since many thermal sizes could not be given over the whole temperature range because no data were available. However, the deformation behavior was in good agreement with the laboratory tests. A correction factor of 0.1367 was iteratively determined, where the deviation of the simulation from the laboratory experiment was less than 0.5%. The deformation simulated after calibration is shown as a dot stroke line and almost coincided with the laboratory experimental results. All other data refer to the calibrated simulation.

The simulated temperature curve during the manufacturing process is shown in Figure 10 in which the maximum and average temperatures are plotted for the individual steps. The peak points of the maximum curve indicated the individual exposures since the melting temperature was present at these times. The times in between corresponded to the cooling between the individual exposures. The course of the average temperature decreased at the beginning, which was due to the growing component volume. The increase in the later course, where even the maximum temperature no longer rose to the manufacturing temperature, was due to the manufacturing of the cantilever arm of the component. This had a significantly larger area to expose, which meant that more heat was supplied to the component, and the component generally heated up. After completing the buildup process, both temperature curves dropped to the ambient temperature, i.e., the manufacturing process was complete.

Figure 11 represents the simulated distortion bridge, in which the equivalent stress, according to von Mises, and the deformation before (a) and after (b) the removal of the support material are shown. The support material was fully discharged in the simulation to mimic the section shown in Figure 4. It clearly shows how residual stresses were relieved by deformation.

The simulated and measured axial residual stresses were compared to validate the simulations, as shown in Figure 12. The solid and dashed lines represent the measured and simulated results, respectively. As can be seen, the simulated ones were in the same order of magnitude as the measured ones. However, quite large deviations could be seen on the surface due to the hole-drilling nature that could not measure a very fine gradation on the surface.

The simulated microstructural analyses are shown in the following. For this purpose, a representative volume element (RVE) was simulated with an edge length of 0.5 mm, as shown in Figure 13a,b. In Figure 13a, fine grains, which are typical for the SLM process with a significant expansion in the buildup direction (*Z*-axis), can be seen. An average circle-equivalent diameter of 19 ± 11 μm was measured. Grain orientations, which depend on their angle to their reference plane, are presented in Figure 13b. While the longitudinal planes (XZ and YZ) had predominantly flat angles, the angles at the transverse plane (XY) tended to be right-angled. In Figure 14, the frequency of grain orientation with respect to the plane is presented, showing that the grains in the XZ and YZ planes tended to have shallow angles relative to their respective planes. Accordingly, the grains departed from a random orientation and favored a preferred direction in the Z-direction. The orientation of the grains in the XY plane showed an even more pronounced preference for the Z-direction. This indicated that the grains had a preferred direction in the assembly direction (*Z*-axis), which demonstrated the typical anisotropy of the SLM process.

## 4. Conclusions

This work investigated the mechanical properties, microstructure, and residual stresses of additively processed CuSn10 material using an SLM method with a 100 W laser.

Spherical powder with an average particle size of 17.3 ± 7.3 μm was processed with a laser power of 95 W, a scanning speed of 324 mm/s, and a hatch distance of 0.065 mm. A relative density of 99.77% was achieved. It was possible to determine mechanical–technological characteristic values comparable with the manufacturer’s specifications in the process. A tensile strength of 487 ± 12 MPa with 5 ± 0.5% elongation to rupture, as well as a hardness of 173 ± 3 HV30, were determined. Based on the results obtained, the following conclusions could be drawn:It was found that CuSn10 could be processed well at a laser power of 100 W. However, relatively small layer thicknesses and track spacings were required, resulting in a significant increase in the production time. Accordingly, a more powerful laser unit is recommended for economical use.A simulation model was developed and validated for predicting deformations and residual stresses in Ansys. For this purpose, thermal and mechanical calculations were coupled, and some simplifications were introduced to achieve an acceptable compromise between computation time and imaging accuracy. A calibration factor had to be used to adjust the simulation because the deformations calculated in the thermal–mechanical approach were too large. The necessity of a calibration factor is assumed to be the limitations of the material and simulation model. On the one hand, the material parameters, which were not entirely defined via the process temperature, led to deviations between the simulation and reality. In addition, the method used to describe the relationship between stress and strain offered only a low mapping accuracy beyond the elastic range. This was considered acceptable as only small stresses were expected. Partially high-stress peaks up to 500 MPa were due to singularities in the transition area between base plate and component. No adjustment was made, as these were clearly identifiable and occurred in a non-critical range. Finally, the simplifications in the simulation model led to deviations. However, since the losses could be compensated by introducing a correction factor, leading to a significant time saving, the method was evaluated positively. However, the determined correction value was only valid for the material used; the correction factor is not valid with the used material. The measurement of residual stresses seemed to show outliers. Due to the production and measurement wall, no additional measurements could be carried out. Alternatives to the measurement carried out should be examined, and, if necessary, semi-destructive methods should be used [32,33].The simulated residual stresses showed a comparable intensity and course compared with the measured ones. However, a deviation could be observed on the surface due to the inability of the hole drilling technique in measuring fine gradation on the surface.In a further simulation, the microstructure of the SLM-produced CuSn10 material was modeled. It was possible to simulate the microstructure with relatively small grains (average circle-equivalent diameter of 19 ± 11 μm) in a clear preferential direction. The calculated microstructure corresponds to the expectations and is plausible but could not be validated. The software developed by Ansys only offers a limited range of setting options, which is why an alternative will be used in the long term.

## Figures and Tables

**Figure 1 materials-15-03902-f001:**
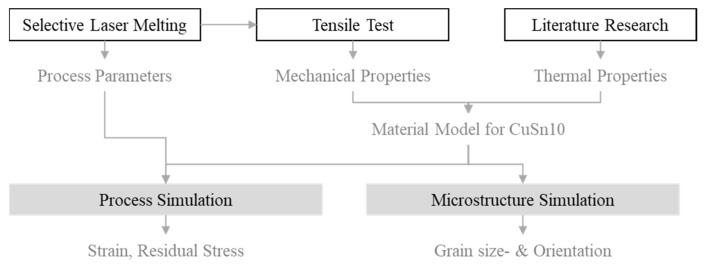
Schematic representation of the experimental plan.

**Figure 2 materials-15-03902-f002:**
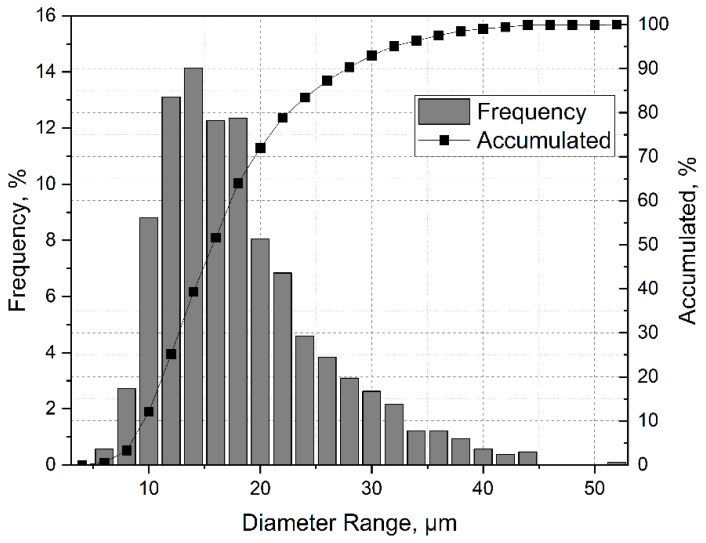
Distribution of powder particle sizes.

**Figure 3 materials-15-03902-f003:**
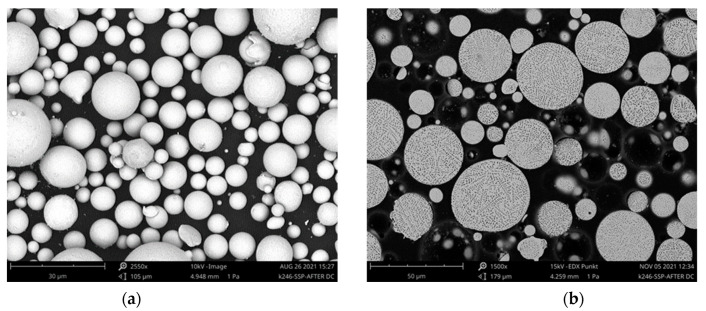
SEM images of the powder used: (**a**) the powder as delivered and (**b**) the microstructure of the particles.

**Figure 4 materials-15-03902-f004:**
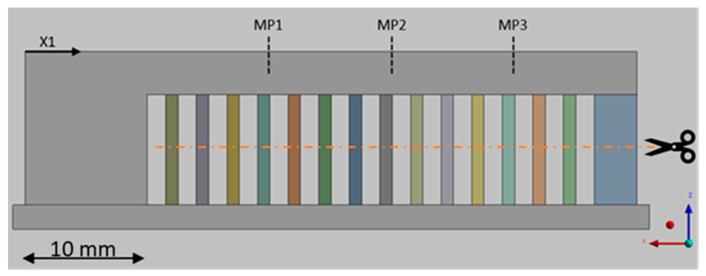
Side view of the distortion bridge with drawn-in measuring ranges for the residual stress.

**Figure 5 materials-15-03902-f005:**
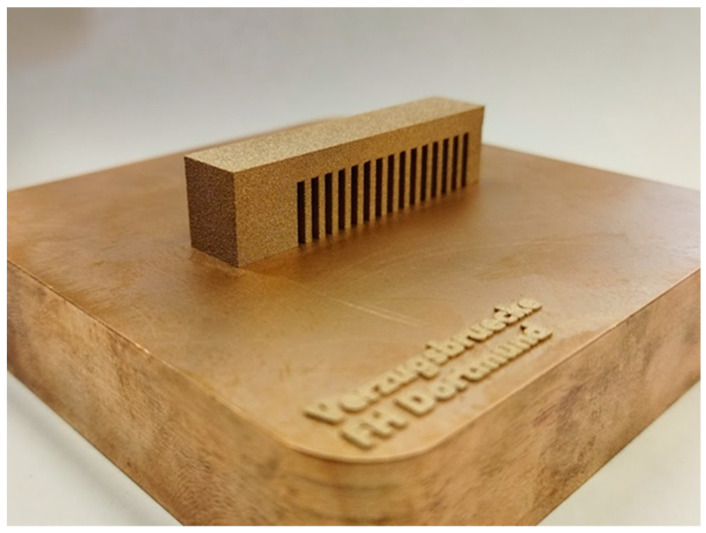
Produced distortion bridge on the building panel.

**Figure 6 materials-15-03902-f006:**
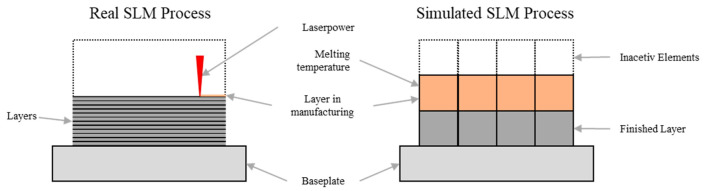
Real and simulated SLM setup in comparison.

**Figure 7 materials-15-03902-f007:**
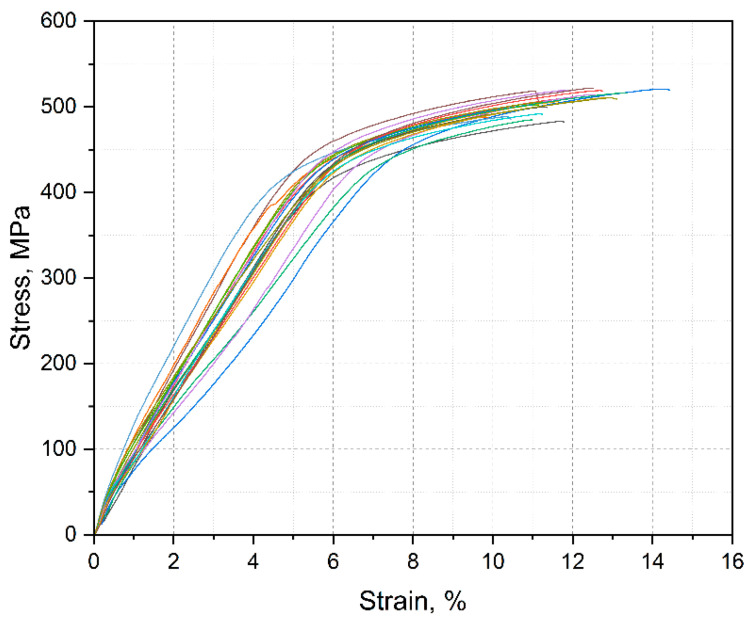
Stress–strain diagram of the examined specimens.

**Figure 8 materials-15-03902-f008:**
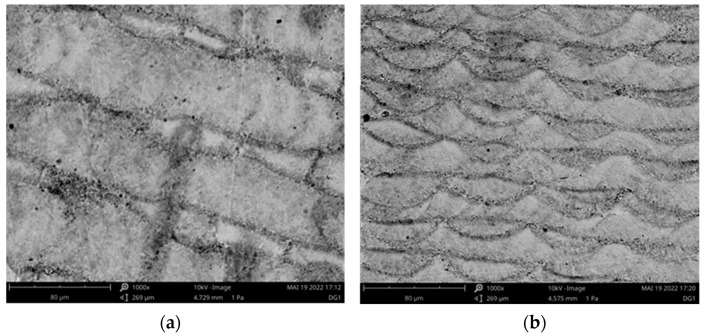
SEM images of the metallographically prepared (**a**) transverse and (**b**) longitudinal cross-sections of a manufactured sample.

**Figure 9 materials-15-03902-f009:**
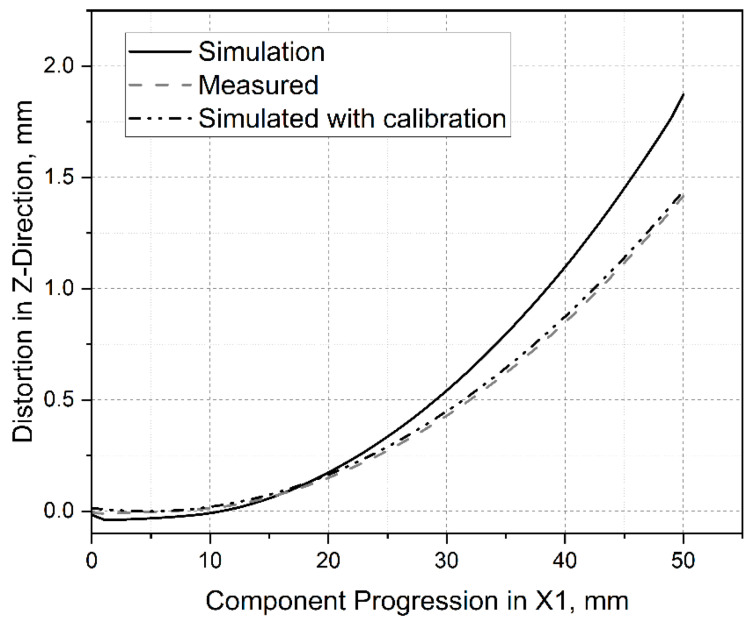
Measured and simulated deformations over the measuring section X1 in the laboratory test.

**Figure 10 materials-15-03902-f010:**
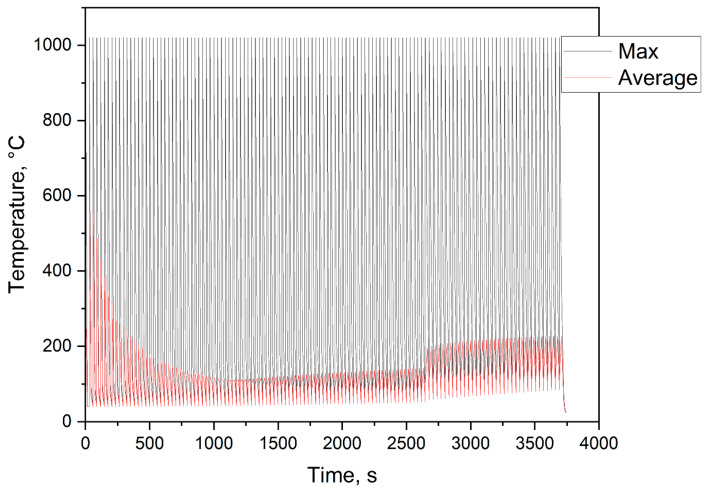
Thermal curve during the simulated production.

**Figure 11 materials-15-03902-f011:**
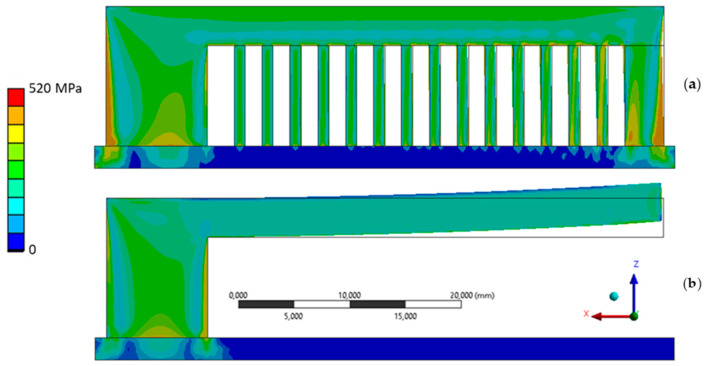
Simulated equivalent stress according to von Mises (**a**) before and (**b**) after the removal of the support material.

**Figure 12 materials-15-03902-f012:**
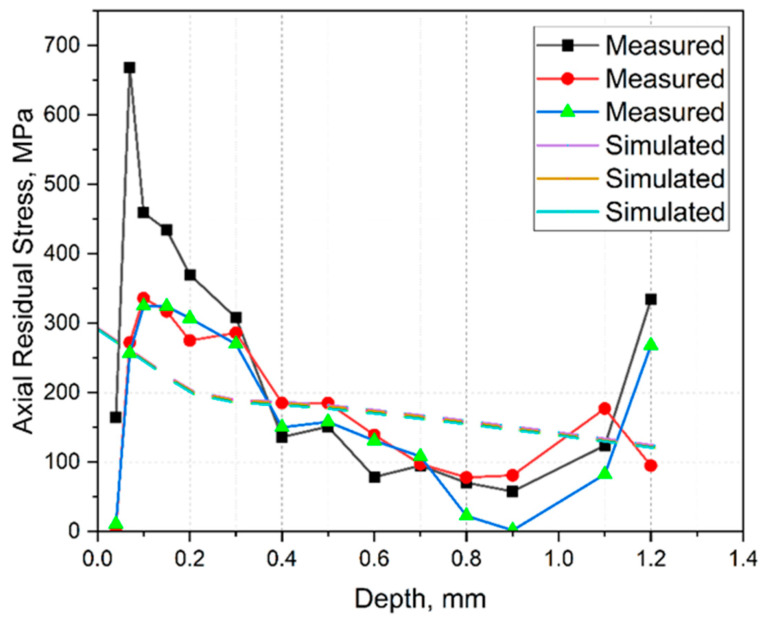
Comparison between the simulated and measured residual stress.

**Figure 13 materials-15-03902-f013:**
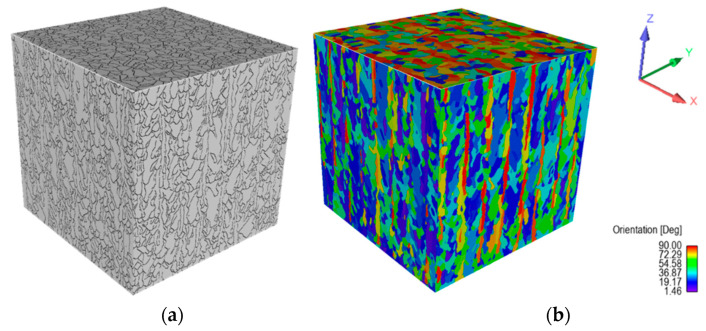
Simulated cube with an edge length of 0.5 mm showing (**a**) the grain boundaries and (**b**) the grain orientation.

**Figure 14 materials-15-03902-f014:**
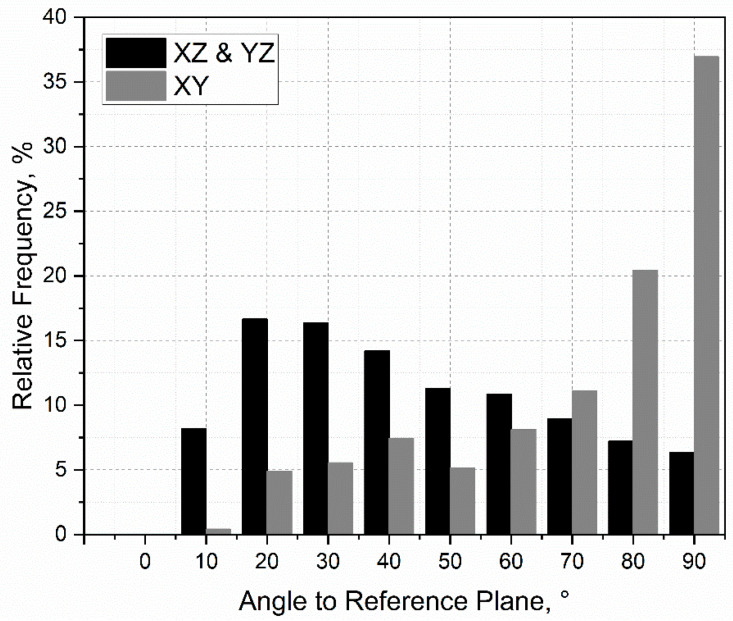
Orientation of the grains relative to the referential plane.

**Table 1 materials-15-03902-t001:** Parameters used for manufacturing the samples.

Parameter	Numerical Value	Unit
Laser power	95	W
Scan speed	324	mm/s
Hatching	0.065	mm
Layer thickness	0.02	mm
Inert gas	Nitrogen	

**Table 2 materials-15-03902-t002:** Parameters used for developing the simulation model.

Parameter	Numerical Value and Unit	Temperature (°C)	Reference
Density	8.76 g/cm³	20	Measured
Coefficient of thermal expansion	0.0000193 1/K	20	[26]
Liquidus temperature	1020 °C	-	[26]
Yield strength	420 MPa	20	Measured
Modulus of elasticity	102 GPa	20	
	100 GPa	100	
	96 GPa	200	[26]
	92 GPa	300	
	87 GPa	400	
Coefficient of thermal conductivity	59 W/(m × K)	20	
	67 W/(m × K)	100	[26]
	76 W/(m × K)	200	
Specific heat capacity	0.38 J/(g × K)	20	
	0.40 J/(g × K)	800	
	1.00 J/(g × K)	850	[27]
	1.65 J/(g × K)	1000	
	0.40 J/(g × K)	1020	

**Table 3 materials-15-03902-t003:** Setup for the process simulation.

Parameter	Numerical Value	Unit	Reference
Coating time	9.5	s	Measured
Preheating temperature	22	°C	No preheating
Gas and powder temperature	22	°C	No preheating
Process temperature	40	°C	Adapted
Gas and powder convection coefficient	0.00001	W/(mm² × K)	

**Table 4 materials-15-03902-t004:** Parameters used for microstructure simulation.

Parameter	Numerical Value	Unit	Reference
Cooling rate	308,100	K/s	Simulated
Temperature gradient	3,258,504	K/m	Simulated
Melt track width	0.083	mm	Measured
Melt track depth	0.03	mm	Measured

**Table 5 materials-15-03902-t005:** Overview of the determined mechanical–technological properties.

Reference	Density (g/cm³)	Yield Strength (MPa)	Tensile Strength (MPa)	Elongation (%)	Hardness HV
Supplier [31]	-	-	430	7	170
Experiment	8.76	420 ± 124	487 ± 12	5 ± 0.5	173 ± 3

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
