# Peer review of "Selective Laser Melting of CuSn10: Simulation of Mechanical Properties, Microstructure, and Residual Stresses"

_materials, 2022, doi:10.3390/ma15113902_

Round 1

Reviewer 1 Report

1. During the introduction part, the authors describe that the main purpose of the present research is to combine different methods to study the material properties. However, the connections between each method require detailed explanations. 

2. According to the authors, the material parameters used in the simulation process are taken from relevant literature. Please quote the literature and explain the viability of these parameters in the present research works, since the parameter used in other literature may not be able to describe the material in the present study correctly. 

3. The simulation process shown in Figure 7 is different from the true SLM fabricating process (track-by-track and layer-by-layer). Have the authors considered the influences that this difference brings to the study?

4. Please highlight the relevance between the tensile properties and the simulation works.

5. Please compare the details of the simulation results shown in Figure 12 with the fabricated section shown in Figure 5.

6. Please indicate the detailed principle to establish the RVE, and also explain the connection between the RVE model and the study. 

Reviewer 2 Report

Review report on the topic ‘Selective Laser Melting of CuSn10: Simulation of Mechanical Properties, Microstructure, and Residual Stresses’. Comments are listed below:

  1. Add the key conclusion of the works in the last two lines of the abstract section.
  2. Discuss the motive behind the work. The clear application of the work should be discussed in the introduction section. Also discuss the novelty of the work in a separate section.
  3. There are numerous spelling and grammatical errors. Please revise the manuscript thoroughly. Sentences are also not complete and references are also cited in a rough manner.
  4. Try to make a bridge between current and previously published work and specify the gap area and objective of the work. There is no discussion related to the residual stress measurement and major works published on residual stresses. Refer to the following works: https://doi.org/10.1016/j.matlet.2020.128347; https://doi.org/10.1016/j.measurement.2014.12.047.
  5. The introduction section is very lengthy. Remove the unnecessary information.
  6. Mention the equation references.
  7. Also, discuss the experimental section in detail. Provide the image of the laser melted surface.
  8. Provide the image of the tested specimen and related results in a separate Table.
  9. The image quality (Fig. 9) is very poor. Replace it with good quality SE image.
  10. Provide a separate section of the discussion.
  11. Simulation work needs more clear discussion.
  12. In present, it looks like a technical report. The results are presented without any technical discussion.
  13. Add key bullet points in the conclusion section instead of the paragraph.

Author Response

Please see the attachment (the response to both reviewers can be found in the attachment)

Round 2

Reviewer 1 Report

The paper has been carefully revised.

Reviewer 2 Report

Accepted